# Clinical Risk Factors for Uterine Cervical Elongation among Women with Pelvic Organ Prolapse

**DOI:** 10.3390/ijerph18179255

**Published:** 2021-09-02

**Authors:** Yi-Yin Liu, Chiu-Lin Wang, Zi-Xi Loo, Kun-Ling Lin, Cheng-Yu Long

**Affiliations:** 1Department of Obstetrics and Gynecology, Kaohsiung Municipal Siaogang Hospital, Kaohsiung Medical University, Kaohsiung 81267, Taiwan; 1060593@kmuh.org.tw (Y.-Y.L.); 990363@kmuh.org.tw (C.-L.W.); 2Department of Obstetrics and Gynecology, Kaohsiung Medical University Hospital, Kaohsiung Medical University, Kaohsiung 80756, Taiwan; 3Department of Obstetrics and Gynecology, Kaohsiung Municipal Ta-Tung Hospital, Kaohsiung Medical University, Kaohsiung 80145, Taiwan; 1030394@kmuh.org.tw (Z.-X.L.); nancylin95@gmail.com (K.-L.L.)

**Keywords:** cervical elongation, hysteropexy, pelvic organ prolapse, predictor, risk factor

## Abstract

Background: Cervical elongation is commonly associated with pelvic organ prolapse (POP). It was an identified risk for recurrent prolapse after hysteropexy, requiring additional surgeries. The aim of the study is to investigate the risk factors for uterine cervical elongation among women with POP. Methods: In this single-center retrospective cohort study, women who underwent vaginal total hysterectomy for POP between 2014 and 2016 were collected. The cervical and total uterine lengths were measured by pathologists, while the ratio of cervical length to total uterine length were calculated. The cervical elongation is defined as corpus/cervix ratio ≤ 1.5. Results: A total of 133 patients were enrolled in this study. Among these patients, 43 women had cervical elongation and 90 women had normal length of uterine cervix. We found that age > 65 years old (67.4% vs. 42.2%, *p* = 0.007), total vaginal length ≥ 9.5 cm (65.1% vs. 45.6%, *p* = 0.035), uterine weight < 51 gm (72.1% vs. 52.2%, *p* = 0.03), and Pelvic Organ Prolapse Distress Inventory 6 (POPDI-6) ≥ 12 (30.2% vs. 14.4%, *p* = 0.032) were associated with the risk of cervical elongation. There were no significant differences on preoperative urodynamic parameters in the two groups. Conclusion: The patient age > 65 years old, the total vaginal length of POP-Q system ≥ 9.5 cm, uterine weight < 51 g, and POPDI-6 ≥ 12 are independent risk factors of cervical elongation in women with POP. For women scheduled for pelvic reconstructive hysteropexy, concomitant cervical amputation should be considered.

## 1. Introduction

With the increase in life expectancy over the past century, pelvic organ prolapse (POP) is a growing gynecologic problem. The lifetime risk of requiring at least one operation to correct incontinence or prolapse is estimated to be approximately 11% [1]. During the past decade, uterine-preserving prolapse surgeries for treating female POP gradually increased in Taiwan [2]. The gold standard of uterine-preserving prolapse surgery is sacrohysteropexy, whereas several novel surgeries, such as uterus-preserving laparoscopic lateral suspension with mesh [3] and laparoscopic long mesh surgery with augmented round ligaments [4], also had promising results on treating female apical prolapse.

A recent systematic review with meta-analysis shows uterine-preserving prolapse surgeries improve operating time, blood loss, and risk of mesh exposure compared with similar procedures with concomitant hysterectomy, and there are no significant differences of short-term prolapse recurrence risk [5,6]. However, advanced uterine prolapse and lack of surgical experience were two significant predictors of POP recurrence following transvaginal hysteropexy surgery in our previous study [7].

Cervical elongation is a well-known problem commonly associated with POP. A single-center prospective, case-control study shows women with symptomatic POP have significantly higher ratios of cervical length to total uterine length than women without POP [8]. Besides, previous case series suggested that cervical elongation can develop or re-develop after uterus-preserving surgery for POP, and may require further surgical intervention [9,10]. However, further randomized studies are warranted for confirmation of these observations.

As cervical elongation may lead to the recurrence of POP after uterus-preserving surgery, it is crucial to identify the risk factors of cervical elongation in women with POP, which can help us decide preoperatively which patient may need concomitant cervical amputation. Therefore, the study aims to identify the predictors of cervical elongation in women with POP.

## 2. Material and Methods

This was a retrospective cohort study conducted in the Urogynecology Department of Kaohsiung Medical University Hospital between January 2014 and December 2016. We enrolled one hundred and fifty-six women with POP stage II to IV, defined by the POP quantification (POP-Q) staging system [11]. They underwent vaginal hysterectomy with or without transvaginal mesh (TVM) procedures at our hospital. Concomitant midurethral sling operations, including tension-free vaginal tape (TVT) (Gynecare TVT, Ethicon, Inc., Piscataway, NJ, USA), TVT-O (Gynecare TVT-Obturator System, Ethicon, Inc., Somerville, NJ, USA), Monarc (AMS, Inc., Minnetonka, MN, USA), and MiniArc (AMS, Inc., Minnetonka, MN, USA), were performed in women with current or occult urodynamic stress incontinence (USI), unless patient did not desire additional surgeries. All women performed preoperative evaluations including pelvic examination, pelvic ultrasonography, multichannel urodynamic studies, and a personal interview to evaluate overactive bladder symptom score (OABSS), the short forms of Urogenital Distress Inventory (UDI-6), and the Pelvic Organ Prolapse Distress Inventory 6 (POPDI-6). Twenty-three patients were excluded due to incomplete medical records. The remaining 133 women were included for statistical analysis.

After hysterectomy, the specimen was measured by pathologist, including the uterine weight, uterine corpus length, and cervical length. The corpus/cervix ratio (CCR) was calculated. Several criteria have been proposed to define cervical elongation [12,13,14,15,16]. Our definition of cervical elongation adopted by the study of A.R. Mothes et al. [16], in which cervical elongation was classified as physiological (grade 0, CCR > 1.5), grade I (CCR > 1 and ≤1.5), grade II (CCR > 0.5 and ≤1), and grade III (CCR ≤ 0.5). In our study, patients with CCR ≤ 1.5 were considered clinically significant and recruited for further analysis [16].

Ethics approval by the institutional review board of our hospital was obtained for retrospective data analysis. The analysis of the potential risk factors involved in cervical elongation included the demographics, stage of POP, involved compartment, uterine weight, urinary symptoms, and questionnaires, as well as preoperative urodynamic parameters. Statistical analysis was performed using SPSS version 19.0 for Windows (SPSS Inc., Chicago, IL, USA). The data of demographic characters are presented as number (percentage), mean (standard deviation). The categorical variables were analysis by chi-square or Fisher’s exact test. In addition, logistic regression model was used to assess the independent predictive value of the variables. A difference was considered statistically significant when *p* was <0.05.

## 3. Results

A total of 133 patients were enrolled in this study, the mean age of the patients was 66.4 ± 8.7 years old, mean parity was 3.3 ± 1.3, and all of the patients were postmenopausal. Six patients had previous POP and/or SUI surgeries. Twenty-nine (21.8%) patients had concomitant midurethral sling surgery (Table 1).

Among all subjects, 43 women had cervical elongation and 90 women had normal cervical lengths. The group with cervical elongation had significantly more women aged 65 or older (67.4% vs. 42.2%, *p* = 0.007), with total vaginal length ≥ 9.5 cm (65.1% vs. 45.6%, *p* = 0.035), uterine weight < 51 gm (72.1% vs. 52.2%, *p* = 0.03), and POPDI-6 ≥ 12 (30.2% vs. 14.4%, *p* = 0.032) compared with the group with normal length of uterine cervix (Figure 1), whereas there were no significant differences in parity, overweight (BMI ≥ 24), prolapse stage, and preoperative lower urinary tract symptoms (Table 2). Besides, between the two groups, there were no significant differences on preoperative urodynamic parameters including detrusor overactivity, maximum flow rate, residual urine, maximum vesical capacity, detrusor pressure at peak flow, functional urethral length, and maximum urethral closure pressure (Table 3).

The logistic regression was performed on the age > 65, total vaginal length ≥ 9.5 cm, uterine weight < 51 gm, and POPDI-6 ≥ 12 to predict the cervical elongation. The *p* values were all with significance (*p* < 0.001, *p* = 0.015, *p* = 0.01, *p* = 0.006, respectively) (Figure 2).

## 4. Discussion

Nosti et al. found that patients with cervical elongation were more likely to be older [15]. In our study, patient age older than 65 years was a predictor of cervical elongation, which was similar to the previous study. In contrast, other studies did not find the same association. Ibeanu et al. reported that age was not a risk factor for cervical elongation and Berger et al. reported that age was inversely related to cervical elongation [12,14]. The difference among studies may be explained by the different definitions of cervical elongation and study populations. Ibeanu and colleagues clinically defined cervical elongation as cervical length measure by POP-Q C-D ≥ 8 cm [14]. Berger et al. use MRI study to define a uterine corpus length of 63 mm and cervical length of 33.8 mm (CCR = 1.9) or cervix to corpus ratio ≤ 0.79 (CCR = 1.26) as normal [12]. Our definition of uterine cervical elongation was based on a CCR ≤ 1.5, which was measured by the post-hysterectomy uterine specimen.

The value of preoperative POP-Q measurements or ultrasound estimates of uterine cervical length is controversial. The POP-Q examination estimate of cervical length correlates fairly with pathologic measurement (r = 0.3, *p* = 0.005); transvaginal ultrasound measures of cervical length did not appear to correlate with anatomic (r = 0.19, *p* = 0.14) or examined cervical length by POP-Q (r = −0.13, *p* = 0.18) [17]. In contrast, Finamore et al. compared preoperative POP-Q points C minus D to actual cervical length recorded in the pathology report. The result showed significant difference between estimated cervical length and actual cervical length (5.6 cm ± 2.91 vs. 3.2 cm ± 0.99, *p* < 0.0001) [13]. Berger et al. found that women with cervix > 33.8 mm were significantly younger than women with normal cervical length. However, this result did not find significant differences in patients with cervix/corpus ratio > 0.79 compared to women with normal ratio [12]. Therefore, the discrepancy between studies is mainly due to the different definitions of the cervical elongation.

In addition, the mean age of study population in our study is older than those studies. Advancing age is a risk factor for POP; there was a 10% increased risk of POP for each decade of life [18]. There were significant reductions in uterine size and in the corpus to cervix ratio after menopause. The reduction in uterine size was related to the number of postmenopausal years [19]. It supports that advancing age may be correlated with cervical elongation. In our study, we also found that uterine weight < 51 gm was a predictor to cervical elongation. There was a positive correlation between the estimated uterine volume and actual uterine weight [20]. Since the uterine size and uterine volume were decreasing with age, the uterine weight might also decrease with age.

Advanced prolapse has been proposed to be a predictor for cervical elongation [16,21]. However, we did not find association between prolapse stage and cervical elongation in our study. Similarly, Berger et al. showed that POP-Q point C, not the stage of prolapse, is a predictor of cervical elongation [12]. In our study, most cases were POP-Q stage III and IV; only 11 cases were POP-Q stage II. The correlation between prolapse stage and cervical elongation may not be proved in the small sample size. But we observed significantly higher POPDI-6 scores in the group with cervical elongation compared with the other group. The POPDI-6 scores significantly correlated with the prolapse stage [22]. It implied that the women with a more advanced stage of prolapse had more distress symptoms, which may be associated with cervical elongation.

Vaginal length was 24% longer in women with prolapse as previously reported [23]. In our study, women with TVL over 9.5 cm were more common in the group of cervical elongation. A previous observational cross-sectional study reported TVL seems to be a confounder in the relationship between cervical station and prolapse symptoms [24]. It implies that the patient with longer vaginal length has more advanced POP and will be exposed to the risk of cervical elongation.

Bladder outlet obstruction is a common urodynamic finding among women with advanced POP [25,26]. Approximately 21% of women with stage II POP and 33% of those with stage III or IV POP report difficulty of bladder emptying [27]. To our knowledge, this study is the first to examine cervical elongation and its association with urodynamic parameters. However, no correlation between cervical elongation and urodynamic parameters was identified. It is suggested that cervical elongation did not disturb the voiding function. A previous case series study in women with cervical elongation following sacrospinous hysteropexy noted that most patients have minimal symptoms, even in those sexually active. Only one out of five women required partial trachelectomy due to bothersome symptoms of vaginal protrusion [9].

Cervical elongation can pose surgical challenges on vaginal hysterectomies and may increase operative time [15]. In addition, it may cause a higher risk of POP recurrence after hysteropexy surgery [9,28] and require further surgical intervention. There was lack of studies to determine cervical elongation as a relative contraindication for uterus-preserving pelvic reconstruction surgery. Further studies are needed to investigate whether cervical elongation occurred as a result of POP alone, regardless of surgical interventions, or whether it represents a local reaction to the introduction of synthetic mesh adjacent to the cervix. To identify women with risk of cervical elongation is important for patients who prepare to undergo uterine preservation surgery for POP.

The limitation of this study is that it is a retrospective study. We examined the cervical elongation by post-hysterectomy specimen. It is difficult to evaluate the weight of the uterus before surgery. We cannot find the correlation between the preoperative ultrasound image and pathologic reports. Further prospective study is needed. To evaluate whether the cervical elongation truly negatively impacts on uterus-preserving surgery, better organized study is needed.

## 5. Conclusions

In conclusion, we found that women aged over 65, total vaginal length over 9.5 cm, uterine weight less than 51 gm, and POPDI-6 scores over 12 are four independent risk factors of cervical elongation in women with POP. Preferably, concomitant trachelectomy can be considered if these women are scheduled for uterus-preserving pelvic reconstructive surgery for POP.

## Figures and Tables

**Figure 1 ijerph-18-09255-f001:**
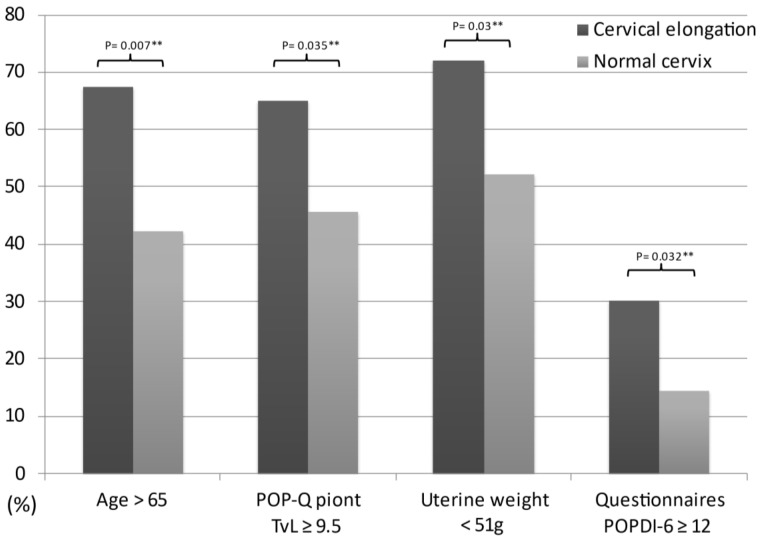
The risk factors which had statistical significant difference. ** means significant difference.

**Figure 2 ijerph-18-09255-f002:**
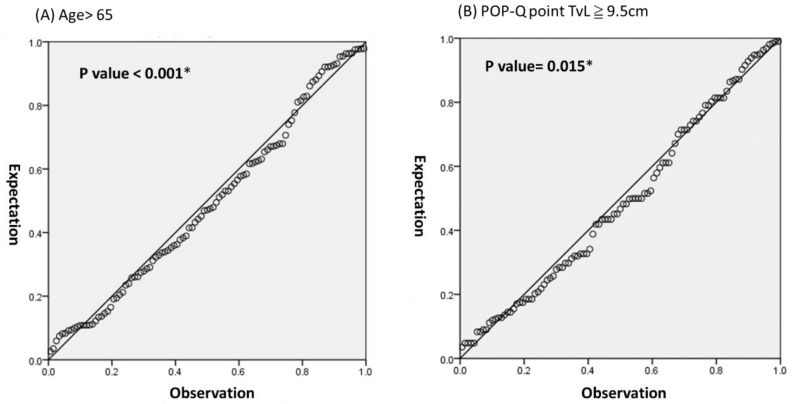
The logistic regression model, (**A**) age > 65, (**B**) POP-Q point TvL ≧ 9.5 cm, (**C**) uterine weight < 51 g, (**D**) POPDI-6 ≥ 12; * Statistical significance.

**Table 1 ijerph-18-09255-t001:** Demographic characteristics of women (*n* = 133) with pelvic organ prolapse undergoing vaginal hysterectomy.

Mean age (years)	66.4 ± 8.7
Mean parity	3.3 ± 1.3
Mean BMI (kg/m^2^)	24.2 ± 3.2
Menopause	133 (100)
Current hormone therapy	23 (17.3)
Current smokers	1 (0.8)
Diabetes mellitus	24 (18.1)
Hypertension	55 (41.4)
History of POP and/or SUI Surgery	6 (4.5)
Concomitant mid-urethral sling suegry	29 (21.8)

Data are given as mean ± standard deviation or *n* (%). BMI, body mass index; POP, pelvic organ prolapse; SUI, stress urinary incontinence.

**Table 2 ijerph-18-09255-t002:** Analysis of clinical features in both groups.

		Cervical Elongation (*n* = 43)	Normal Cervix (*n* = 90)	OR (95% CI)	*p* Value
Age	≤65	14 (32.6)	52 (57.8)		
	>65	29 (67.4)	38 (42.2)	3.88 (1.13–13.29)	0.007 **
Parity	<3	10 (23.3)	25 (27.8)		
	≥3	33 (76.7)	65 (72.2)	0.65 (0.13–3.19)	0.58
BMI healthy	<24	24 (55.8)	51 (56.7)		
Over weight	≥24	19 (44.2)	39 (43.3)	0.85 (0.26–2.82)	0.93
Past history	H/T	18 (41.9)	37 (41.1)	0.85 (0.26–2.82)	0.94
	HT	6 (14.0)	17 (18.9)	0.70 (0.36–2.70)	0.48
	DM	6 (14.0)	18 (20.0)	0.97 (0.24–3.88)	0.4
	Previous POP or SUI surgery	0	6 (6.7)		0.18 *
Prolapse stage	Ⅱ	4 (9.3)	7 (7.8)		
	Ⅲ–Ⅳ	39 (90.7)	83 (92.2)	1.11 (0.13–9.63)	0.75 *
POP-Q point	Ba ≥ 3	27 (62.8)	45 (50.0)	1.69 (0.22–2.96)	0.17
	C ≥ 3	21 (48.8)	37 (41.1)	2.41(0.32–8.86)	0.4
	TvL ≥ 9.5	28 (65.1)	41 (45.6)	2.23(1.12–11.2)	0.035 **
Involved compartment	Anterior wall	37 (86.1)	76 (84.4)	1.14(0.23–3.12)	0.81
	Uterine prolapse	41 (95.3)	80 (88.9)	0.69 (0.21–2.29)	0.34
	Posterior wall	8 (18.6)	16 (17.8)	0.80 (0.09–6.86)	0.91
Uterine weight	<51 g	31 (72.1)	47 (52.2)	2.83(1.13–12.2)	0.03 **
Preop symptoms	Frequency	23 (53.5)	45 (50.0)	0.93 (0.27–3.12)	0.71
	SUI	12 (27.9)	37 (41.1)	0.44 (0.11–1.74)	0.14
	UI	18 (41.9)	44 (48.9)	0.78 (0.23–2.61)	0.45
	Incomplete emptying	38 (88.4)	70 (77.8)	2.31 (0.28–19.10)	0.14
	Hesitancy	36 (83.7)	63 (70.0)	4.40 (0.54–35.70)	0.09
Questionnaires	OABSS ≥ 10	6 (14.0)	15 (16.7)	2.77 (0.74–10.36)	0.69
	UDI-6 ≥ 6	21 (48.8)	36 (40.0)	1.23 (0.37–4.07)	0.34
	POPDI-6 ≥ 12	13 (30.2)	13 (14.4)	4.13 (1.16–14.72)	0.032 **

Data are given as *n* (%). BMI, body mass index; H/T, hypertension; HT, hormone therapy; DM, diabetes mellitus; POP, pelvic organ prolapse; SUI, stress urinary incontinence; Preop, preoperative; UI, urgency incontinence. * Fisher’s exact test; ** Statistical significance.

**Table 3 ijerph-18-09255-t003:** Comparison of preoperative urodynamic parameters in both groups.

		Cervical Elongation (*n* = 43)	Normal Cervix (*n* = 90)	OR (95% CI)	*p* Value *
DO		10 (23.3)	18 (20.0)	1.21 (0.32–3.88)	0.67
Q max (mL/s)	<15	33 (76.1)	67 (74.4)	1.23 (0.37–4.07)	0.77
	≧15	10 (23.3)	23 (25.6)		
RU (mL)	<50	18 (41.9)	49 (43.3)		
	≧50	25 (58.1)	51 (56.7)	2.41 (0.71–8.16)	0.87
MCC (mL)	<350	7 (16.3)	16 (17.8)	0.54 (0.15–1.92)	0.83
	≧350	36 (83.7)	74 (82.8)		
Pdet (cm H_2_O)	<15	2 (4.7)	3 (3.3)	2.93 (0.68–12.65)	0.66 *
	≧15	41 (95.3)	87 (96.7)		
FUL (mm)	<25	6 (14.0)	15 (16.7)	1.13 (0.34–3.75)	0.69
	≧25	37 (86.0)	75 (83.3)		
MUCP (cm H_2_O)	<40	7 (16.3)	77 (7.8)	0.97 (0.24–3.88)	0.14
	≧40	36 (83.7)	81 (92.2)		

Data are given as *n* (%). DO, detrusor overactivity; Qmax, maximum flow rate; RU, residual urine; MCC, maximum cystometric capacity; Pdet, detrusor pressure at peak flow; FUL, functional urethral length; MUCP, maximum urethral closure pressure. * Fisher’s exact test.

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
