# Peer review of "Clinical Risk Factors for Uterine Cervical Elongation among Women with Pelvic Organ Prolapse"

_ijerph, 2021, doi:10.3390/ijerph18179255_

Round 1
Reviewer 1 Report
Comments and Suggestions for Authors
This in an interesting manuscript aiming to identify possible predictors of cervical elongation in women with POP. Although the methods are adequately described, the result section could be improved and maybe presented more extensively. Nonetheless, after a minor revision this paper should be considered for publication.
Author Response
Thank you for the comment, pleased see the attachment of revision articles, we had improved the part of result section.

Reviewer 2 Report
This was a single-center, retrospective study that sought to evaluate factors associated with cervical elongation in women who underwent vaginal total hysterectomy for pelvic organ prolapse (POP). The goal was to identify women who may need to have concomitant cervical amputation, however the authors did not explain possible benefits to such a procedure even if these women could be correctly identified preoperatively. My comments follow:
- The authors incorrectly identify this study as a retrospective case-control study (where cases with a certain condition and control without the condition) are identified and then evaluated for exposures. This was a retrospective cohort study as women were selected based on an exposure (vaginal total hysterectomy for POP) and then evaluated for the condition of interest (cervical elongation).
- Please explain the possible benefits of concomitant cervical amputation in women with cervical elongation, and therefore the need to identify women with cervical elongation prior to POP surgery.
- Please cite the statistical software package used for analyses.
- There is no reason not to perform Fisher’s exact tests for all categorical variables as most statistical software packages have this option.
- Identifying risk factors (i.e. association) is not synonymous with identifying factors that are predictive. These are different types of analyses. The authors identified risk factors but did not explore the whether the presence or absence of these factors predicted the presence of cervical elongation.
- The authors stated they identified independent risk factors for cervical elongation but it is not clear whether they did this type of analyses. Logistic regression was used, but it is not clear whether multivariable logistic regression modeling was done. This is necessary to determine whether a factor is independently associated with an outcome. If multivariable regression was done, please provide more details for the criteria for entry and retention of variables in the final model.
- The results in Table 2 are incorrectly displayed in a variety of ways:
- There are odds ratios and p-values displayed for every category. This is methodologically incorrect. Categories for the variables should be mutually exclusive and one should be the referent category and therefore the OR should be 1 for this categorically (traditionally displayed in tables as referent). Odds ratios are then calculated based on comparison to the referent category. Similarly, there should only be one p-value per categorical variable if it is from chi-square or Fishers. If using the p-value from logistic regression (wald chi square test for example) then p-values are displayed for the categories that are not the referent. I do not intend to be disrespectful to whomever analyzed your data, but I suggest consulting an experienced biostatistician.
- The odds ratios and p-value for age must be reversed in the table as the authors repeatedly state that older age is associated with cervical elongation.
- The confidence intervals and p-values frequently contradict each other. If the 95% CI includes one, then the p-value should be >.05. If the p-value is <.05 then the CI should not include 1.
- It is not clear whether the ORs and Cis displayed are unadjusted or adjusted. If they are adjusted, please add a footnote as the variables included in the model. I would suggest including only the unadjusted results in table 2 and make a separate table for a final multivariable model. Alternatively, move the current unadjusted p-value column to the left of ORs and display the adjusted ORs with an additional column with p-values form multivariable logistic regression.
- If the ORs are unadjusted, why are they not displayed for all variables?
- P-values for are not provided for some of the variables. Please include univariate p-values at minimum.
- Table 3 seems to lack all the issues noted for Table 2. It is still unclear whether the ORs are unadjusted or adjusted as they are not given for all variables. If they are adjusted, please add a footnote as he variables that were included in the model.
- The discussion is disjointed and lengthy. For instance, the authors discuss age then definition s of cervical elongation then POPQ then back to age. I suggest restructuring and editing for conciseness.
Reviewer 3 Report
The work presented to me for review concerns the analysis of predictors of cervical elongation. I agree with the authors that after operations involving the disturbance of the statics of the reproductive organ, the cervix lengthens over time. So far, there is not much work dealing with this problem, which significantly increases the value of the work. In the introduction, the authors write only about the uterine suspension surgery, but other operations should also be mentioned, such as suspension of the cervix to the promontory, lateral suspension of the cervix according to Dubuissone or Richter's surgery. This should be completed in the introduction. Materials and methodology written briefly and to the point, legibly. When it comes to presenting the results, I would suggest that the age, cervical elongation, uterine weight and POPDI-6 ≧ 12 should be presented graphically - the most important values ​​would be much clearer. Discussion written succinctly and to the point. Literature from the last 25 years.
In my opinion, the work clearly presents the problem of cervical elongation after surgeries correcting cervical statics. In the summary, the authors provide the reader with specific knowledge - what factors should be taken into account in a patient qualified for surgery due to disturbed cervical statics, which may affect postoperative lengthening of the cervix and possibly perform a simultaneous shortening of the cervix in such a patient.
The work is written legibly, touching on an important aspect in the surgical treatment of a disorder of the statics of the reproductive organ.
Author Response
Thanks for your comment. We improved the section of introduction and add the figure of important values. Please see the revision articles.
Round 2
Reviewer 2 Report
The presentation of the results have improved significantly.
- Since the authors did not use logistic regression, what method was used to calculate odds ratios (or relative risks) and 95% confidence intervals?
- Why did the authors not use multivariable logistic regression to determine which of the factors are independent risk factors? I suspect from the results presented that the only independent risk factor is older age. If this is true, I think this is an important finding given that the premise of the study was to try to identify women with cervical elongation pre-operatively. If age is the only independent risk factor, then it is likely not possible to reliably identify these women pre-operatively based on the factors evaluated in this study.
- In the conclusions, line 188, the authors still state that there are four independent risk factors. Again, they did not do the appropriate analyses to determine this. I am skeptical that all four factors stated are independent risk factors given the p-values.
- I omitted this comment from my prior review by mistake, but why did the authors use BMI>=24 as the cutoff for overweight/obesity? The traditional cutoff is >=25. This will not alter the results much but will make them consistent with the vast majority of published studies that evaluate BMI as a risk factor.
Author Response
Dear Reviewer
Thanks for your comment
please see the attachment for the reply.
